# On The Presence of Double-Descent in Deep Reinforcement Learning

Viktor Veselý[1], Aleksandar Todorov[1], and Matthia Sabatelli[*1]

[1]University of Groningen
[1]m.sabatelli@rug.nl

## 1 Abstract

The double descent (DD) paradox, where over-parameterized models see generalization improve past the interpolation point, remains largely unexplored in the non-stationary domain of Deep Reinforcement Learning (DRL). We present preliminary evidence that DD exists in model-free DRL, investigating it systematically across varying model capacity using the Actor-Critic framework. We rely on an information-theoretic metric, Policy Entropy, to measure policy uncertainty throughout training. Preliminary results show a clear epoch-wise DD curve; the policy's entrance into the second descent region correlates with a sustained, significant reduction in Policy Entropy. This entropic decay suggests that over-parameterization acts as an implicit regularizer, guiding the policy towards robust, flatter minima in the loss landscape. These findings establish DD as a factor in DRL and provide an information-based mechanism for designing agents that are more general, transferable, and robust.

## 1 Introduction & Motivation

The classical machine learning theory posits a U-shaped bias-variance curve, suggesting that excessive model complexity leads to overfitting and degraded generalization. The discovery of Double Descent (DD) [1] has fundamentally challenged this, showing that test error can fall again as model capacity increases far beyond the point of perfect fit (the interpolation threshold). While DD has been widely observed within the Supervised Learning (SL) regime [2], its presence in Deep Reinforcement Learning (DRL) has, to the best of our knowledge, not yet been characterized. We argue that the reason for this gap is that studying DD in DRL is particularly complex. Unlike static supervised tasks, DRL involves a non-stationary environment where the data distribution (experience) changes with the evolving policy. Crucially, there are no predefined training and validation sets that can be used to observe and characterize the generalization properties of the phenomenon. Furthermore, typical DRL losses are not reliable indicators of generalization. Let us take as an example, the classic Mean Squared Error (MSE) loss used by a variety of value-based and actor-critic algorithms for the value function ($V$):

$$L_V(\theta_v) = \frac{1}{2}\mathbb{E}\left[(r_t + \gamma V(s_{t+1}; \theta_v) - V(s_t; \theta_v))^2\right].$$

In this expression, $\theta_v$ represents the value network parameters; $r_t$ is the instantaneous reward; $\gamma \in [0, 1]$ is the discount factor; $V(s_{t+1}; \theta_v)$ is the estimated value of the next state (the bootstrap target); and $V(s_t; \theta_v)$ is the current state value estimation. Contrary to the Supervised Learning case, where high losses are usually correlated with poor learning performance, this Temporal Difference (TD) loss can increase throughout training and also be a sign of successful learning. As the loss rises, it often signifies that the agent is learning to bootstrap more effectively with respect to temporal difference targets likely associated with novel states, suggesting a better coverage of the state space underlying the Markov Decision Process. This ambiguity makes typical DRL losses ill-suited for the generalization analysis required to characterize DD. Yet, understanding whether DD governs DRL performance is crucial, as modern agents overwhelmingly rely on vast, over-parameterized neural networks [3]. This phenomenon is particularly interesting to study because of its practical implications: DRL agents currently suffer from poor generalization [4], exhibit a loss of plasticity [5], and rarely transfer well between tasks [6]. Therefore, there is much to gain if one could quantify how general a DRL agent can be. We aim to establish the presence of DD in model-free DRL and identify the key dynamics accompanying it, thereby linking the phenomenon to the core problem of generalization in DRL.

## 2 Experimental Approach

To investigate DD, we employed the Actor-Critic (A2C) algorithm on the popular `Frozen-Lake` environment. We define model complexity by varying the hidden layer width and depth (capacity) of the shared policy/value backbone, exploring configurations such as $[64]$, $[64, 64]$, $[128, 128]$, $[64, 64, 64]$, and $[128, 128, 128]$ following approaches alike to Supervised Learning [2]. The core of our analysis lies in

---

*Corresponding Author.

tracking the dynamics of policy uncertainty during training, using the information-theoretic metric of Policy Entropy as our primary lens. This metric measures the randomness of the action distribution, providing insights into the policy's confidence and exploration throughout the learning process. For a given policy $\pi(\cdot|s)$ in state $s$, the policy entropy is defined as:

$$\mathcal{H}(\pi) = -\mathbb{E}_\pi\left[\log\pi(a|s)\right] = -\sum_{a\in\mathcal{A}}\pi(a|s)\log\pi(a|s).$$

We track this entropy across training episodes to establish the correlation between policy uncertainty and the onset of the second descent.

## 3    Preliminary Findings

Our results provide initial evidence for the existence of the Double Descent phenomenon in Deep Reinforcement Learning. Our preliminary findings, detailed in Figure 1, illustrate this effect. The graph plots the average Policy Entropy, $\mathcal{H}(\pi)$, as a function of training episodes for five distinct actor-critic network architectures. Policy Entropy $\mathcal{H}(\pi)$ is the core metric, quantifying the uncertainty of the agent's action selection. We can see that as training progresses, agents across the capacity spectrum exhibit the classic DD curve: models that pass the interpolation point enter a regime where generalization initially degrades (first descent) but then dramatically improves (second descent) as training continues. This transition to the second descent is strongly coupled with the dynamics of policy uncertainty. Specifically, agents entering the second descent display a consistent and sustained decrease in Policy Entropy. This decrease indicates that the over-parameterized models are efficiently pruning uncertainty and settling on highly deterministic, high-information policies that maximize reward. The dynamics of this entropic compression are strongly capacity-dependent. Small capacity models (e.g., [64] and [64, 64]) demonstrate rapid convergence to near-zero entropy within 2,000 episodes, indicating policy fixation on a locally optimal solution. In contrast, the characteristic entropy signature associated with the Double Descent phenomenon is clearly visible in the highly over-parameterized networks (purple: [128, 128, 128]), which maintain significantly higher average entropy and larger variance for an extended duration. Furthermore, the [64, 64, 64] capacity (red line) exhibits an intriguing multi-phase descent pattern, suggesting a prolonged, high-entropy exploration followed by a re-ascent (around episode 3,500) before its final descent, a behavior sometimes associated with Triple Descent in highly complex, over-parameterized models [7]. This delayed descent in uncertainty is theorized to be the result of implicit regularization,

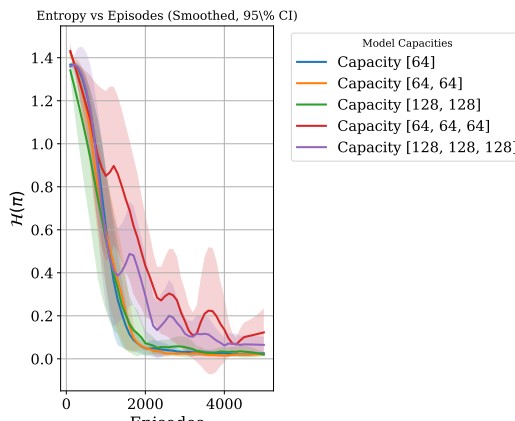

**Figure 1.** Policy Entropy, $\mathcal{H}(\pi)$, as a function of training episodes across five distinct actor-critic network architectures. The shaded regions represent the 95% confidence interval (CI) over fifteen independent runs.

guiding the models toward flatter, more generalizable minima. We posit that this sustained entropic compression reflects this implicit regularization effect: the over-parameterized system uses its high capacity to discard irrelevant or non-robust policies and converge to a simpler, low-entropy solution corresponding to a flat, highly-generalizing minimum in the objective landscape. The shaded regions in the graph represent the 95% confidence interval (CI) across fifteen independent experimental runs.

## 4    Future Implications

These initial results demonstrate that the Double Descent phenomenon extends to model-free DRL, confirming that over-parameterized agents use redundancy to find better solutions rather than simply overfitting. Our results seem to suggest that this performance is a direct result of implicit regularization guiding agents toward deterministic policies in flat, robust minima. This finding has profound implications, but we must formally evaluate whether the policy stability and entropic compression observed here translate to superior out-of-distribution (OOD) generalization. As a crucial next step, we will evaluate whether there is a significant difference in generalization for models that exhibit Double Descent. We aim to test this on OOD tasks using the procedurally generated environments introduced by Cobbe et al. [8] to rigorously assess the models' ability to transfer learned skills to unseen environment variations. Understanding the dynamics between capacity and policy uncertainty could lead to novel, explicit regularization techniques, pushing agents more reliably into the second descent regime for highly general, transferable, and flexible DRL systems.

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
