# OpenReview forum: "On The Presence of Double-Descent in Deep Reinforcement Learning"
_NLDL.org/2026/Abstracts_Track — NLDL 2026 Abstracts_

### Official Review · Reviewer_BRW1 · 2025-10-24

**Soundness:** 4
**Correctness:** 3
**Rating:** 5
**Confidence:** 4

**Summary:**

This work shows preliminary evidence of double descent in model-free Deep Reinforcement Learning: varying Actor-Critic model capacity reveals a clear double descent curve, where over-parameterization reduces policy entropy and acts as an implicit regularizer, helping agents leverage redundancy to find robust solutions rather than overfit.

**Strengths:**

1. Novel contribution to the research community
2. Very well-written paper, with clear explanations
3. Convincing preliminary findings
4. Clear future work directions

**Weaknesses:**

Experimental results are currently limited, and need comparisons to related works and  evaluation across additional datasets/ frameworks.

---

### Official Review · Reviewer_WRNL · 2025-10-26

**Soundness:** 3
**Correctness:** 3
**Rating:** 4
**Confidence:** 3

**Summary:**

This paper investigates the presence of the Double Descent (DD) phenomenon in Deep Reinforcement Learning. Their preliminary findings indicate a DD like curve in policy entropy, after initial degradation in generalization (first descent), overparameterized models recover (second descent) with reduced entropy, suggesting more deterministic and robust policies.

**Strengths:**

1. The paper explores a largely uncharted area by extending double descent framework to the reinforcement learning domain.
2. The motivation is well articulated.
3. The authors described the setup transparently.
4. The forward-looking discussion was good and indicate a mature research trajectory.

**Weaknesses:**

1. Study made in a single toy environment, which may not exhibit sufficient complexity for strong generalization claims.
2. The argument that “overparameterization acts as implicit regularization” is plausible but unproven within the presented scope. It requires ablation or theoretical grounding.
3. Policy entropy is an indirect proxy for generalization; it captures uncertainty but not necessarily the agent’s performance on unseen states. No test-time validation or OOD metric is presented.
4. The paper’s insights are primarily conceptual at this stage, and its implications for real-world DRL systems remain to be demonstrated.

These are all recommendations for the extended version of this work :)

Good luck!

---

### Official Review · Reviewer_yQtW · 2025-11-02

**Soundness:** 3
**Correctness:** 3
**Rating:** 4
**Confidence:** 3

**Summary:**

The abstract presents preliminary results that indicate the presence of the double descent phenomenon in deep reinforcement learning. These results are based on applying the actor-critic (A2C) algorithm to the Frozen-Lake environment, varying the model complexity and tracking the policy entropy across training episodes. Based on these initial findings, the authors propose next steps for further investigations into the phenomenon.

**Strengths:**

- The idea of investigating DD in RDL is clearly and coherently presented.
- The initial findings are interesting and point towards exciting next steps.
- The methodology behind the experiments is clear and sound.

**Weaknesses:**

- The preliminary findings are only based on a single algorithm on a single environment.
- Policy entropy is used as a measure of the models' generalization ability. A clearer justification of why this is a valid measure, would strengthen the presentation of the initial findings.

---

### Decision · Program_Chairs · 2025-11-05

**Decision:**

Accept

**Comment:**

The abstract is of interest to the community and should be presented at the conference.